# Virus-Derived Chemokine Modulating Protein Pre-Treatment Blocks Chemokine–Glycosaminoglycan Interactions and Significantly Reduces Transplant Immune Damage

**DOI:** 10.3390/pathogens11050588

**Published:** 2022-05-16

**Authors:** Isabela R. Zanetti, Michelle Burgin, Liqiang Zhang, Steve T. Yeh, Sriram Ambadapadi, Jacquelyn Kilbourne, Jordan R. Yaron, Kenneth M. Lowe, Juliane Daggett-Vondras, David Fonseca, Ryan Boyd, Dara Wakefield, William Clapp, Efrem Lim, Hao Chen, Alexandra Lucas

**Affiliations:** 1Center for Personalized Diagnostics (CPD), Biodesign Institute, Arizona State University (ASU), Tempe, AZ 85287, USA; izanetti@asu.edu (I.R.Z.); mburgin@asu.edu (M.B.); liqiang.zhang@asu.edu (L.Z.); ram.ambadapadi@asu.edu (S.A.); jyaron@asu.edu (J.R.Y.); david.fonsecaarce@yale.edu (D.F.); 2Ionis Pharmaceuticals, Inc., Carlsbad, CA 92008, USA; syeh@ionisph.com; 3Department of Animal Care and Technologies, Biodesign Institute, Arizona State University (ASU), Tempe, AZ 85287, USA; jacki.kilbourne@asu.edu (J.K.); kenneth.m.lowe@asu.edu (K.M.L.); juliane.daggett@asu.edu (J.D.-V.); 4School for Engineering of Matter, Transport and Energy, Ira A. Fulton Schools of Engineering, Arizona State University, Tempe, AZ 85287, USA; 5Center for Applied Structural Discovery, Biodesign Institute, Arizona State University, Tempe, AZ 85287, USA; rjboyd@asu.edu; 6Pathology Department, University of Florida, Gainesville, FL 32611, USA; dwakefield@ufl.edu (D.W.); clapp@pathology.ufl.edu (W.C.); 7The Biodesign Center of Fundamental and Applied Microbiomics, Center for Evolution and Medicine, School of Life Sciences, Arizona State University, Tempe, AZ 85287, USA; efrem.lim@asu.edu; 8The Department of Tumor Surgery, Second Hospital of Lanzhou University, Lanzhou 730030, China; chenhao3996913@163.com; 9Center for Immunotherapy, Vaccines and Virotherapy (CIVV), Biodesign Institute, Arizona State University, Tempe, AZ 85287, USA

**Keywords:** antisense, chemokine, glycosaminoglycans, inflammation, kidney, M-T7, rejection, transplant, virus

## Abstract

Immune cell invasion after the transplantation of solid organs is directed by chemokines binding to glycosaminoglycans (GAGs), creating gradients that guide immune cell infiltration. Renal transplant is the preferred treatment for end stage renal failure, but organ supply is limited and allografts are often injured during transport, surgery or by cytokine storm in deceased donors. While treatment for adaptive immune responses during rejection is excellent, treatment for early inflammatory damage is less effective. Viruses have developed highly active chemokine inhibitors as a means to evade host responses. The myxoma virus-derived M-T7 protein blocks chemokine: GAG binding. We have investigated M-T7 and also antisense (ASO) as pre-treatments to modify chemokine: GAG interactions to reduce donor organ damage. Immediate pre-treatment of donor kidneys with M-T7 to block chemokine: GAG binding significantly reduced the inflammation and scarring in subcapsular and subcutaneous allografts. Antisense to N-deacetylase N-sulfotransferase1 (ASO^Ndst1^) that modifies heparan sulfate, was less effective with immediate pre-treatment, but reduced scarring and C4d staining with donor pre-treatment for 7 days before transplantation. Grafts with conditional Ndst1 deficiency had reduced inflammation. Local inhibition of chemokine: GAG binding in donor organs immediately prior to transplant provides a new approach to reduce transplant damage and graft loss.

## 1. Introduction

According to the Global Observatory on Donation and Transplantation (GODT) data, surgeries for organ transplantation in 2019 reached a record number with 153,863 transplants, approximately 17.5 transplants per hour, and an increase of 4.8% from 2018 (http://www.transplant-observatory.org/, acessed on 1 January 2022) [1]. Renal transplants comprise a large share of the transplanted organs. Diabetic and hypertensive renal disease are responsible for the majority of cases, with kidney failure requiring dialysis and/or transplantation [1,2]. In 2018 only 36.2% of 95,479 renal transplants came from living donors, the majority of transplants worldwide are obtained from deceased donors (http://www.transplant-observatory.org/, accessed on 1 January 2022) and, among those, patients with severe brain damage [2]. Transplanted organs are damaged even prior to engrafting, by ischemia, low blood flow, during transport or shock, by surgical trauma and immunological damage induced by cytokine storm in donors with severe brain injury [3,4]. Thus, even before surgery, the donated organ is subjected to excess immune cell activation and damage. This early inflammation and damage can cause long term functional impairment [3,4,5], in addition to the acute and recurrent episodes of antibody-mediated rejection. Early damage is linked to late progressive vasculopathy and scarring in allografts, termed chronic rejection. Recent studies in rodent models have demonstrated progressive inflammatory immune cell activation through cell adhesion molecule expression, causing an influx of leukocytes in the kidneys of brain-dead rats, when compared to non-brain-dead controls [3,4]. Severe brain injury is also associated with increased cytokine and chemokine activity, associated with immune cell activation. Increased expression of chemokines and chemokine receptors is detected in brain death and associated with endothelial dysfunction [3,4,5,6].

Viruses have developed highly effective immune modulators over millions of years of evolution; agents designed to evade the host’s immune response to infections. We have developed virus-derived immune-modulating proteins as a new class of therapeutic. M-T7 is a 37 KDa myxoma virus-derived purified protein that binds C, CC and CXC classes of chemokines, blocking chemokine to GAG binding [7,8,9,10,11]. In prior studies, modifying chemokine and chemokine receptor activity reduced transplant damage and rejection [8,9,10,11,12,13,14]. Treatment after transplant with M-T7 reduced the aortic allograft vascular inflammation at 30 days follow-up [8]. M-T7 also reduced inflammation and scarring and improved long term renal allograft survival in mouse and rat models in separate studies [9,10,14]. Modifying the endothelial polysaccharides and GAG composition also reduced immune cell response and organ damage in a separate series of transplant models, including aortic and renal allograft studies. [9,10,11,12,13,14,15,16,17,18,19].

In prior work, engrafted renal allografts reduced early histological markers of allograft rejection with either M-T7 treatment, starting on the day of transplant, or in allograft kidneys from mice with conditional N-deacetylase-N-sulfotransferase-1 (Ndst1) deficiency, further supporting a central role for chemokine: GAG interaction in transplant rejection [9]. Ndst1 is a sulfotransferase enzyme that modifies heparan sulfate (HS), the predominate GAG component in the endothelial glycocalyx. HS GAG binds the chemokines directing chemokine gradient formation and cellular invasion. Donor renal allografts from Ndst1^−/−^ donor mice had significantly reduced scores for cell infiltrates, vasculitis, glomerulitis and tubulitis in renal allografts when compared to saline-treated control grafts [9]. No additional treatment was given to these mice after transplantation, suggesting that interfering with chemokine to HS GAG binding in donor allografts is capable of reducing early graft inflammation, reducing immune cell activation, invasion and organ damage.

Chemokines are chemoattractant cytokines that direct immune cells to sites of injury in donor organs and increase immune cell activation [5,6,7,8]. Inflammation and cellular activation in donor organs can be protective but, when excessive, have the potential to cause progressive damage to donor transplants with scarring, vasculitis and graft loss [5]. Chemokines are small proteins classified into four subgroups (C, CC, CXC and CX_3_C), as defined by the number of amino acids (X) separating the two N terminal cysteine residues. CC, CXC and CXC chemokines and their receptors are increased after transplant with evidence for rejection [5,6,20]. Chemokines act locally to direct leukocyte adhesion, extravasation and navigation along the GAG gradients in the connective tissue glycocalyx surrounding the endothelium during immune response activation, forming gradients that direct leukocyte traffic or invasion at sites of tissue injury [5,6]. Chemokine to receptor interactions are promiscuous, but the combination of chemokine binding to specific receptors, as well as binding to GAGs in the glycocalyx, governs the tissue immune responses. The glycocalyx layer is composed of polysaccharides that include GAGs such as heparan and chondroitin sulfate. Together with connective tissue proteins, the glycocalyx covers the luminal surface of vascular endothelial cells. Interactions between chemokines and GAGs thus allow immune cell attachment and migration [14,21,22] and direct immune cell migration into engrafted organs [21,22].

The majority of current treatments for transplants are focused on decreasing immune T cell and antibody-mediated rejection after engrafting, e.g., treatment of the transplant recipient after surgery, to prevent rejection [23]. Immune cell activation and invasion, when excessive during overactive inflammation or rejection, can damage transplanted organs prior to transplantation, causing damage to the donor’s organs even before harvest for transplantation, and can lead to graft loss. While current immune modulating therapeutics are highly effective for acute antibody-mediated rejection, this treatment is less effective for late or chronic rejection, that occurs after the first year post transplant, increasing graft loss. Treatment for late or chronic rejection remains less effective, necessitating repeat transplantation in some patients or a return to hemodialysis. There is also a risk of anti-rejection medication induced toxicity [23]. High doses of immunosuppressants such as calcineurin inhibitors and corticosteroids increase the susceptibility to opportunistic infections amongst other severe complications, including malignancies, diabetes and Cushing’s syndrome. New drugs are now under investigation as approaches to reduce this excess damaging inflammation in donor grafts [23,24,25,26,27,28,29,30].

Treatment of grafts prior to surgical transplant is postulated to improve graft outcomes by decreasing inflammation and organ damage before transplantation [10,11,12,25,26,27,28,29,30,31,32,33,34]. Dopamine pre-treatment of donor organs has demonstrated reduced renal allograft rejection and superior long-term graft survival [24]. Similar beneficial findings have been observed with pre-treatment of brain-dead donor vascular composite allografts (VCAs) with CR2-Crry, a targeted complement inhibitor, that blocks C3 activation and the generation of biologically active complement, specifically C3 opsonins, C3a, C5a and the membrane attack complex. C3a and C5a are involved in endothelial activation and immune cell recruitment [31]. Pre-coating of grafts with CR2-Crry ameliorated the ischemia-reperfusion injury (IRI) in grafts from brain-dead donors and prolonged graft survival in animal models [31]. Recent studies have investigated pre-treatment of liver and heart transplants with small interfering RNA (siRNA) approaches with different targets, including caspase 3 and complement factor C5a, in preclinical models [32,33].

In summary, chemokine: GAG interactions are predicted to drive damaging immune cell invasion in engrafted organs. We have postulated that treatment of renal allografts prior to engrafting, e.g. pre-treatment with therapeutics designed to interfere with chemokine: GAG interactions, will reduce early donor graft damage and rejection (Figure 1). Here we investigate pre-treatment (PT) with both M-T7, as well as an antisense (ASO) construct targeting Ndst1 (ASO^Ndst1^) prior to donor organ implantation. Pretreatment of donor kidneys was given either immediately prior to engrafting with soaking for 1 h before subcapsular allograft implant (PTS), or by pre-treatments given for 7 days to the donor mouse before organ resection and transplant (7dsPT) (Figure 2). Subcapsular renal as well as subcutaneous transplants were both examined after PTS (soaking) pre-treatment. Our studies indicate that interfering with the chemokine: GAG axis immediately prior to transplantation can reduce early transplant immune damage.

## 2. Results

### 2.1. Soaking Pre-Treatment (PTS) of Subcapsular Renal Allografts with M-T7 Significantly Reduced Inflammation at 15 Days Follow Up 

At 15 days follow-up, the areas and diameters of inflammation in the engrafted sections pretreated by soaking with M-T7, were significantly reduced when compared to Saline PTS (Figure 3A–E; *p* < 0.0003). Micrographs of renal allograft implants at 15 days are shown in Figure 3, panels A–D. An independent histopathology score analyzing tubulitis, glomerulitis, scarring and a combined histopathology score detected similar significant reductions with M-T7 treatment at 15 days follow-up in grafts pre-treated for one hour by PTS (Figure 3, Panels G,H) immediately prior to transplantation.

Renal allograft subcapsular implants pretreated with M-T7 by soaking (PTS) for one hour prior to subcapsular transplant produced a trend toward a decrease in inflammation at 3 days follow-up, but this trend did not reach significance (Figure 3E; *p* = NS). Histopathology scores for glomerulitis (Figure 3G, *p* < 0.0188 ANOVA) and the overall histopathology score (Figure 3H, *p* < 0.0002 ANOVA) were significantly reduced after M-T7 PTS at 15 days. ASO^Ndst1^ did reduce inflammation at 3 and 15 days in comparison to saline (*p* < 0.0001 ANOVA; *p* < 0.0403 for 3 days follow-up and *p* <0.0003 at 15 days follow-up). However, the ASO^Ndst1^ effects at 15 days were equal to the control ASO^Scr^ (*p* = 0.9763) suggesting a non-specific effect for ASO^Ndst1^ on inflammation in PTS treated grafts (Figure 3D,F).

At 3 and 15 days follow-up, a reduction in the measured area for scarring in allografts was detected with the M-T7 PTS treatment. M-T7 reduced scarring both as a measured area (Figure 4A,B) and as a histopathology score (Figure 4C, ANOVA *p* < 0.0302). Tubulitis scores were not significantly changed by any treatment (*p* = NS, not shown). ASO^Ndst1^ reduced measured inflammation at 15 days follow-up in the PTS model (*p* < 0.0403, Figure 3F) but did not have a detected significant reduction on pathology scoring for glomerulitis (Figure 3G) or combined score (Figure 4C). ASO^Ndst1^ PTS treatment did demonstrate a decrease in histopathology score for scarring (*p* < 0.0425, Figure 4C), but did not demonstrate a significant decrease in measured scar areas (Figure 4A). A trend towards an increased number of detected glomeruli with intact morphometry was seen in M-T7 PTS treatments, but did not reach significance (Figure 4D, ANOVA *p* < 0.5437; *p* = NS).

Implant of kidney allografts derived from conditional Ndst1 knockout (Ndst1^−/−^) mice also significantly reduced inflammation and scarring at 3 days post-transplant (Figure 3F, *p* < 0.0403). These findings are consistent with prior work where transplant of whole functioning Ndst1^−/−^ transplants (renal allografts derived from conditional Ndst1 knock out mice) into BALB/c mice produced a significant reduction in acute rejection [9]. 

ASO^Ndst1^ PTS reduced the independent histopathology scores for scarring (Figure 4C, *p* < 0.0425), but neither for glomerulitis (*p* = 0.6820) nor for the overall histopathology score (*p* = 0.0689), when compared to saline or ASO^Scr^ control PTS treatments (Figure 3G). For glomerulitis, ASO^Scr^ treatment was similar to saline treatment, suggesting that this histopathology scoring assessment also demonstrated a nonspecific effect for ASO^Ndst1^ and ASO^Scr^ treatments. 

No adverse effects were detected with either treatment approach, M-T7 or ASO^Ndst1^, with neither increased mortality nor infections, as seen in prior work [7,8,9,14,35,36]. Overall, the mortality for subcapsular transplants was 80%, with no significant differences for any of the treatments, whether given as PTS or as 7ds PT.

### 2.2. Immunohistochemical Analyses of Immune Cell Invasion

The only treatment given to allografts in the PTS or 7ds PT models was given prior to engrafting, with no follow-up treatment of recipient mice after engrafting. Early effects on immune cell responses were therefore assessed at 3 days after transplantation. F4/80+ Macrophage infiltrates were significantly reduced by M-T7 treatment at 3 days follow-up in the subcapsular PTS model (ANOVA *p* < 0.0008) (Figure 5A). 

M-T7 PTS soaking significantly reduced the macrophage invasion (*p* < 0.0006), with reductions similar to the reductions seen in Ndst1^−/−^ conditional knock out kidney transplants when compared to Saline PTS (*p* < 0.0009) (Figure 5A,E,F). In contrast, ASO^Ndst1^ did not alter macrophage invasion (*p* = 0.2007). CD3 positive (CD3+) T cell (Figure 5B,G,H; *p* = 0.1994) and Ly6G+ neutrophil counts (Figure 5C; *p* = 0.0684) in each graft implant were not significantly altered by any of the pre-treatments. Ndst1^−/−^ implants at 3 days follow-up had reduced neutrophil counts when compared to Saline PTS treatment of WT C57BL/6 kidney implants, while M-T7 and ASO^Ndst1^ trended toward a non-significant increase (Figure 5C). CD19 B cell counts were significantly decreased (ANOVA *p* < 0.0164) by the ASO^Ndst1^ pre-treatment of C57BL/6 mice (*p* < 0.0131) and in the Ndst1^−/−^ allograft implants (*p* < 0.0400) (Figure 5D). M-T7 and saline treatments had equivalent effects on the CD19 B cell counts indicating that M-T7 produced neither detrimental suppression of, nor an increase in, B cells, (Figure 5D,I,J). These observations suggest that M-T7 predominantly reduced macrophage invasion in the immediate pre-treatment PTS model.

### 2.3. Seven Day Pretreatment (7dsPT) with M-T7 or ASO^Ndst1^ Did Not Reduce Inflammation, but Did Reduce Scarring

Seven days of pre-treatment of donor mice with either ASO^Scr^ or ASO^Ndst1^ increased the areas of inflammatory cell invasion when compared to Saline 7dsPT at 3 days follow-up (Figure 6A; *p* < 0.0044 ANOVA). M-T7 7dsPT did not reduce inflammation when compared to Saline 7dsPT at both 3 and 15 days follow-up (Figure 6A,B,G,H). At 15 days follow-up, there was no significant change in the area of inflammation with any treatment, although there was a trend toward increased inflammation with ASO^Ndst1^ (ANOVA *p* = 0.2771) (Figure 6B). ASO^Scr^ 7dsPT at 3 days follow-up led to increased inflammation when compared to M-T7 (*p* < 0.0026) but not in comparison to ASO^Ndst1^ 7dsPT (*p* = 0.1433). Scarring was significantly reduced by the pre-treatment of donor mice with M-T7 (*p* < 0.0010), and by ASO^Ndst1^ 7dsPT (*p* = 0.13144) at 3 days follow-up, when compared to saline or ASO^Scr^ controls (Figure 6C, ANOVA *p* < 0.0033). At 15 days follow-up, there was again no overall decrease with 7dsPT with either ASO^Ndst1^ (*p* = 0.3690) or M-T7 (*p* = 0.4517) when compared to saline pre-treatments of donor mice for 7 days prior to transplant (*p* < 0.0556, ANOVA). ASO^Scr^ treatment did increase scarring and was significantly greater than ASO^Ndst1^ (*p* < 0.0479) when given as 7dsPT at 15 days follow-up.

CD3+ T cell and F4/80 macrophage counts (Figure 6E,F) had only borderline decreases on IHC analysis, paralleling the lack of marked efficacy in reducing inflammatory cell invasion when donor mice were pretreated for 7 days prior to graft harvesting and transplant, in contrast to the soaking PTS approach. M-T7 and ASO^Ndst1^ had nonsignificant trends toward reducing F4/80+ cell counts (*p* = 0.1434, Figure 6E) with trends toward reduced macrophage counts for M-T7, ASO^Ndst1^ as well as ASO^Scr^ treatments. ASO^Scr^ pre-treatment increased CD3+ T cell counts when compared to Saline, M-T7 or ASO^Ndst1^ pre-treatments (ANOVA *p* = 0.0135, Figure 6F). M-T7 and ASO^Ndst1^ did not significantly increase or decrease F4/80 macrophage or CD3 positive T cells with 7dsPT when compared to saline (Figure 6E,F).

These findings would suggest that immediate pre-treatment with M-T7 for 1 h prior to allograft transplant was more effective than pre-treatment for 7 days (7dsPT) for preventing inflammation or scarring in renal allografts.

### 2.4. C4d Positive Staining Is Increased in Both the Subcapsular Allografts as Well as the Recipient Kidney

As a secondary analysis for rejection, immunohistochemical staining was performed for the detection of C4d (complement). Areas of dense C4d positive tubules were detected in both the renal allograft implants and in the recipient kidneys with both the PTS and 7dsPT approaches (Figure 7). While subcapsular renal allografts pre-treated for one hour (PTS) with M-T7 before transplantation had significantly decreased inflammation at 15 days follow-up (*p* < 0.0066) (Figure 3E), a reduction in scarring was detected in the 7dsPT model (*p* < 0.0033 ANOVA) at 3 days follow-up with M-T7 pre-treatment (Figure 6C), but not at 15 days. C4d staining in the PTS model did not show a significant reduction.

A nonsignificant trend was detected for decreased inflammation in the renal grafts with M-T7 and ASO^Ndst1^ treatment in the 7dsPT model, but scarring was reduced with both treatments at 3 days follow-up (Figure 6C). In contrast, the implanted allograft kidney had significantly decreased C4d positive staining (*p* < 0.0001) at 15 days follow-up (Figure 7A,E,F,G), with either M-T7 (*p* < 0.0201) or ASO^Ndst1^ (*p* < 0.0001) pre-treatments for 7 days. 

ASO^Ndst1^ and M-T7 counts were reduced compared to both ASO^Scr^ and Saline treatment controls. Thus, ASO^Ndst1^ may be more beneficial when given for 7 days to the donor prior to transplantation rather than when given as a soaking pre-treatment for 1 h. Implant of the Ndst1^−/−^ conditional knockout graft was conversely less effective (not shown). 

There was an increase in detectable C4d positive staining in the recipient BALB/c mouse kidney (Figure 7B,D), as well as in the C57Bl/6 allograft (Figure 7D,E) suggesting a possible systemic response after transplant, either induced by surgical injury or rejection. This recipient kidney increase in C4d staining was not altered by either PTS or 7dsPT for any treatments (Figure 7B).

These findings indicate that 7 days pre-treatment (7dsPT) of donor animals may be a more effective approach for reducing the adaptive immune responses, while immediate pre-treatment soaking may be better when treating with local M-T7 inhibition of chemokine: GAG interactions and inflammation. Further, the allograft implant induces a widespread renal and/or systemic response and this response is suppressed by M-T7 PTS or 7ds PT soaking pre-treatments. 

### 2.5. Subcutaneous Renal Allograft Transplant Demonstrated a Significant Reduction in Inflammation and Scarring with M-T7 PTS

To rule out nonspecific effects secondary to the immediate soaking pre-treatment (PTS) with M-T7, we repeated the study in a second allograft transplant model using a subcutaneous renal allograft transplant model. Sections of kidney from C57Bl/6 mice were soaked for one hour with either M-T7, ASO^Ndst1^, ASO^Scr^ or Saline, as for the PTS for subcapsular transplants, and then implanted as subcutaneous transplants. No further treatment was given after engrafting. PTS with M-T7 again significantly reduced the inflammation in renal allograft implants, as seen with the renal subcapsular transplant model (ANOVA *p* < 0.0088; M-T7 *p* < 0.0026; Figure 8A,D,E). Thus, there was reproducible efficacy in a second allograft implant.

ASO^Ndst1^ PTS treatment neither reduced inflammation nor scarring and did not significantly differ from ASO^Scr^ (Figure 8A,B), again indicating a nonspecific effect (*p* = 0.096 for ASO^Ndst1^; *p* < 0.002 for ASO^Scr^). Ndst1^−/−^ deficient mouse renal transplants did significantly reduce inflammation as seen in the subcapsular transplant model (Figure 8A; *p* < 0.0026). 

In contrast to the subcapsular transplant model, none of the treatments significantly reduced scarring after subcutaneous transplant (*p* = 0.0567 ANOVA; Figure 8B), although there was a trend. This difference in the effect on scarring may be due to the reduced vascularity in the dermal implant sites or due to the marked differences in the local intrinsic immune responses in the dermal layers when compared to the kidney capsule, which may or may not be inter-related.

## 3. Discussion

Chemokine: GAG interactions are central to immune cell responses, providing a directional map or gradient for immune cellular invasion at sites of injury [37]. This integral chemokine response is a local response initiating immune cell responses at sites of tissue injury. We have assessed the efficacy of local disruption of chemokine: GAG interactions in donor organs immediately prior to transplant as a new therapeutic approach to interrupting immune responses and inflammation in donor organs. We have examined the blockade of chemokine to GAG binding using both the Myxoma virus-derived chemokine modulating protein, M-T7, as well as modifying the Heparan sulfate GAG composition using ASO^Ndst1^ treatments. We have detected a significant reduction in inflammation after treating donor organs immediately prior to transplant with M-T7, a chemokine-modulating protein that blocks chemokine: GAG interactions. Immediate pre-treatment of the donor organ with M-T7 had greater efficacy than systemic pre-treatment of donors. In contrast, treatment with ASO^Ndst1^ was not effective when given immediately prior to transplantation, but did show efficacy when given to the donor mouse for 7 days prior to transplant, where M-T7 was not as effective. In summary, interruption of chemokines to GAG binding with M-T7 was effective when given immediately prior to transplant, whereas modification of HS GAG composition with ASO^Ndst1^ required prolonged treatments of the donor prior to engrafting.

In prior work we demonstrated that treatment beginning after transplant, using either the chemokine modulating protein M-T7 that blocks chemokine: GAG interactions or the engrafting of Ndst1 deficient (Ndst1^−/−^) mouse donor kidneys, led to significant reductions in early and late rejection [9]. Here, pre-treatment with either M-T7 or ASO^Ndst1^ in donor allografts was assessed as an approach to prevent early damage and ongoing severe inflammation in donor organs. We examined both initial soaking for one hour immediately prior to transplantation and treatment with systemic injections of donor animals for 7 days prior to transplant. As noted, there was a greater benefit with reduced inflammation and scarring (fibrosis) after M-T7 treatment with the immediate pretransplant soaking (PTS), rather than the 7dsPT. M-T7 also provided a greater reduction in inflammation than did ASO^Ndst1^. Given the nature of the chemokine to GAG interaction, which is presumed to be a local GAG to chemokine binding, this would appear to emphasize the central roles of early local chemokine GAG gradients in donor organs in driving immune cell binding and invasion after transplantation.

As a second analysis for pre-treatment, the donor mouse was treated for 7 days prior to harvest of the kidney for transplantation. In these 7dsPT donor organs there was no significant reduction in inflammation, but there was a significant reduction at 3 days follow-up. Both the M-T7 and ASO^Ndst1^ significantly reduced scarring in the grafts (Figure 5) and both were reduced compared to ASO^Scr^ control and saline controls. Interestingly, the C4d marker for rejection was reduced by ASO^Ndst1^ 7ds PT treatment of donors, while M-T7 was not effective. C4d staining was detected in the recipient kidneys remote from the engrafted organs, but there was no significant change with any treatment for this detected C4d marker in the recipient kidneys. The cause for this detected C4d may be simple organ damage from the surgical implant or true rejection, and may represent a systemic response or rather a local injury response via vascular or lymphatic interactions between the graft site and the recipient kidney. The subcapsular and subcutaneous transplants are predicted to have a significant ischemic component, in addition to the expected incompatibility due to mouse strain mismatch (BALB/c and C57BL/6). This will require further study to demonstrate the cause for this recipient reaction.

Prior work demonstrated efficacy in renal and aortic allografts with M-T7 treatment given after transplant by iv or ip injections [9,14]. This efficacy may, however, be due to both the local and systemic immune modulating effects of M-T7 treatments. We have now demonstrated that modulation of chemokine: GAG interactions using M-T7 immediately prior to graft implant, employing either the soaking or PTS method, is superior to pre-treatment of donor animals for 7 days prior to transplantation for reducing inflammation and rejection. We would postulate that the local pre-treatment or soaking PTS approach provides an immediate directed treatment acting on the donor organ. With pre-treatment of the donor mouse for 7 days, the reduced efficacy, if any, was detected in the Subcapsular PTS model. This may be due to lack of effective access of the differing treatments to the donor organ or more rapid clearing. ASO^Ndst1^ treatment did demonstrate reductions in detectable Ndst1 expression in normal mice. We do not, however, know the treatment distribution of M-T7 in the donor mice treated systemically. ASO^Ndst1^ also reduced scarring as well as C4d staining when given to the donor mouse for 7 days prior to transplant. ASONdst1 treatment may potentially require longer term application for efficacy or may simply not modify the HS GAG glycocalyx composition in a manner that would effectively reduce immune cell invasion. Future work with functional allograft transplants will be a next stage to assessing pre-treatment with agents that modify chemokine: GAG interactions.

The implant of the Ndst1^−/−^ mouse kidneys, derived from conditional knock out mice, when transplanted using the subcapsular PTS model approaches did demonstrate significant reductions in inflammation as well as scarring in the subcapsular PTS model. This efficacy of the Ndst1^−/−^ conditional KO mouse implant for reducing early inflammation and scarring is consistent with an intrinsic benefit for reducing Ndst1 expression in the endothelial cell and myeloid precursor cell populations. This benefit did not, however, translate fully when treating the mouse donor kidneys with PTS or systemic 7ds PT approaches. This reduced benefit with ASO^Ndst1^ pre-treatments may reflect the fact that systemic ASO treatment did not alter graft GAG composition in the same manner as selective endothelial cell conditional knock-out in Ndts1^−/−^ mouse model. The chemokine to GAG interaction is thus a complex multifaceted interaction dependent upon an array of differing chemokines, receptors and GAG species. There is redundancy and overlap of many chemokine interactions with differing cell types. Thus, a non-specific Ndst1 suppression of expression may be less effective or may require longer term treatments after transplantation. Newer ASO constructs are in development by Dr S Yeh and may prove to have greater efficacy.

With each model we examined pre-treatment alone. Neither subsequent immune modulating or immunosuppressant treatment, nor any M-T7 or ASO^Ndts1^ treatment, were given after transplantation. In future work, additional treatments could be given after the initial pre-treatment, as well as potential combined treatment with standard immune suppressants used to prevent rejection. In addition, it should be noted that these are useful models to complete an initial screening for the benefit of pre-treatments, but an analysis of pre-treatment efficacy with and without subsequent ongoing treatments after transplantation should be examined for further efficacy.

To investigate the potential role of modulating chemokine: GAG interactions to reduce renal allograft transplantation we also analyzed two alternative models for testing therapeutics. We have examined both subcutaneous and subcapsular implants as models for renal transplant rejection. Here we demonstrated that both transplant models, kidney to kidney subcapsular and kidney to subcutaneous transplantation, detected significant changes in the generalized inflammatory cell invasion, but scarring was more improved in the subcapsular transplant model after treatment with M-T7. The subcapsular model may provide a better model for transplant rejection than subcutaneous kidney transplantation and provides a closer physiological equivalent to full functional kidney transplant. The two differing recipient mouse implant areas, subcapsular kidney and subcutaneous implants, have known differences in vascularization as well as in local immune responses.

Subcutaneous transplantation has been studied using different types of organ tissue including trachea and hepatocytes [38,39,40], however, although it constitutes an easy and accessible method for surgery, the subcutaneous area is less vascularized than other transplantation sites [41]. What may be of greater relevance is that the skin has a very unique and active immune system differing extensively from many of the internal organs including the kidney [42], which constitutes a great limitation for the success of graft transplant. Studies using subcapsular kidney transplants have presented good results with different organ tissues, including cornea and hepatocytes, the latter with better outcomes specifically in ischemia-reperfusion injury [43,44], contrasting with results in subcutaneous models. Furthermore, the recipient’s microenvironment under the renal capsule is believed to be much more suitable to receive organ tissue than the skin; this supposition is corroborated by results from studies with transplantation of kidney organoids under the kidney capsule resulting in formation of glomerular basement membrane, fenestrated endothelial cells and podocyte foot processes in the absence of any exogenous vascular growth factor [45]. These studies indicate that the use of subcapsular renal transplants provides a simpler allograft transplant model for the study of transplant rejection and vasculopathy as well as new treatment approaches. Full orthotopic kidney transplants in mice requires extensive training in microsurgery, as well as a large number of mice, and the surgeries are technically complex due to their small size. These alternative and simpler approaches in animal models using subcapsular and subcutaneous transplants may offer similar benefits in the evaluation of inflammation and response to treatment, while having lower costs, surgical preparation time and quantity of surgical supplies [46].

Virus-derived immune modulating proteins protect the virus during infection, blocking host immune defenses, especially immune cell invasion. The virus-derived chemokine modulating protein M-T7 reduced renal allograft inflammation and scarring in subcapsular and subcutaneous transplants when given to the donor organ as a pre-treatment immediately prior to transplant. Viruses provide a new approach for investigating mechanisms of transplanted organ rejection and a potential source of new immune modulating protein therapeutics for prevention of transplant organ damage and rejection.

## 4. Methods

### 4.1. Pretreatment of Allografts: Pretreatment Soaking (PTS) of Renal Allografts and 7 Days Pretreatment of Renal Allograft Donors (7dsPT)

All renal allograft procedures were approved by the ASU Institutional Animal Care and Use Committee (IACUC) and conformed to national, international and university guidelines for animal care. We have examined two approaches to pre-treatment of allografts and two approaches for the blockade of chemokine: GAG interactions (Figure 2). Treatment with either M-T7, a chemokine-modulating protein (CMP) that interferes with chemokine: GAG binding as well as pre-treatment with ASO^Ndst1^ were assessed (Table 1, Figure 1). Both soaking pre-treatment/PTS (N = 50 transplants) and 7 days donor pre-treatment/7dsPT (N = 48 transplants) approaches were investigated using treatment with either M-T7 or ASO^Ndst1^ in the subcapsular transplant model. For subcutaneous transplant, only the PTS pre-treatment was examined (N = 62 transplants). As comparators, saline-treated allografts, scrambled sequence ASO (ASO^Scr^) controls or kidneys isolated from a genetic knock out of Ndst1 (Ndst1^−/−^) were assessed in renal allograft implants.

For soaking pre-treatments, PTS, given immediately before transplantation, kidneys from C57BL/6 donor mice after euthanasia were flushed by hand using sterile technique with 200µL of either 1µg/mL M-T7, 25 mg ASO^Ndst1^, 25 mg ASO^Scr^ or saline controls and then placed in oxygenated Dulbecco’s Modified Eagle Medium (DMEM), for one hour prior to transplantation (Figure 2) [32]. For systemic pre-treatment of the donor mice, C57BL6 mice were given intraperitoneal (IP) injections daily for 7 days with 200 µL of either M-T7 protein (1 ng/g body weight) diluted in sterile saline or 25 ng/gm body weight ASO^Ndst1^, or controls, specifically 25 ng/gm ASO^Scr^ or Saline. Ndst1^−^^/−^ renal transplants also served as a non-treated control where recipient BALB/c mice received kidney transplants from mice conditionally deficient in Ndst1 [27]. Renal allografts were treated by PTS for subcapsular and subcutaneous allograft implants. In a second cohort, allografts were also pretreated by giving 7 days pre-treatment to the donor mice prior to subcapsular transplant (7dsPT, Figure 2). No further M-T7, ASO^Ndst1^ or control treatments were given to the recipient BALB/c mice after engrafting.

### 4.2. Subcapsular Renal Allograft Transplant

Mice used included C57BL6/J, BALB/c and N-deacetylase-N-sulfotransferase-1 (Ndst1f/f TekCre^+^ or Ndst1^−/−^) at 8–12 weeks of age (Table 1; N = 50 subcapsular PTS renal transplants, 20 with 3 days follow-up and 28 with 15 days follow-up; N = 48 subcapsular 7dsPT renal transplants, 24 with 3 days follow-up and 24 with 15 days follow-up). Kidneys were isolated after euthanasia from C57Bl/6 mice or Ndst1^−/−^, conditional knock-out mice lacking Ndst1 expression in endothelial cells and myeloid precursors). Sections of C57BL/6 mice kidneys were used as donors for BALB/c recipient mice; One C57BL/6 kidney divided into sections for implant into six recipient BALB/c mice [35]. Sections were cut to incorporate all layers, cortex to hilum. Anesthetized BALB/c mice (Ketamine (100 mg/mL, 120 mg/kg)/xylazine (20 mg/mL, 6 mg/kg) mixture) were shaved to create a 2-inch area on the right side of the mouse over the kidney. The surgical site was cleaned with Chlorhexidine Gluconate 2% solution and 70% ethanol and the surgeries performed in a sterile field. A vertical incision approximately 0.7–1.0 cm in length midway between the bottom of the rib cage and the iliac crest, the kidney exteriorized and a small incision made in the renal capsule. A section of C57BL/6 mouse kidney, including medulla to outer cortex kidney tissue, was then inserted into the subcapsular space and the kidney returned to the abdomen and the surgical site closed with sterile sutures. Mice were given Buprenorphine 0.1 mg per/kg per mouse for pain control and subcutaneous sterile saline (200–500 µL) to aid in recovery. Mice were kept on a heating pad until fully awake and then checked daily for signs of pain and discomfort (hunching, piloerection, loss of appetite, weight loss, etc.). If signs of excess pain were seen, repeat doses of buprenorphine were administered. Mice were monitored by experienced veterinary staff as well as the experimental group. There were two mice lost during transplant, one during subcutaneous implant and one immediately before subcapsular transplant (2% mortality); both were deemed secondary to anesthesia and surgery and were unrelated to specific treatments [35]. No mice were lost after transplant.

### 4.3. Subcutaneous Renal Allograft Transplant

For subcutaneous transplant surgeries, the graft was implanted in a 0.25 inch pocket incision in 1 × 1 inch shaved sections prepared on the back of the mouse between the shoulders [36]. Half of one donor kidney was inserted into the subscapular pocket using sterile forceps and the skin closed with two–three sutures. One mouse provided donor kidney sections for four transplants (Table 1; N = 62 subcutaneous PTS renal transplants, 32 with 3 days follow-up, and 30 with 15 days follow-up). For the subcutaneous transplants only soaking PTS treatments were used. Mice were checked daily for signs of discomfort such as hunching, piloerection and weight loss. After 3 or 15 days post-transplant, mice were euthanized using CO2 gas followed by cervical dislocation.

### 4.4. M-T7 Expression and Purification

M-T7 was expressed in CHO cells in vitro. The Myxoma virus-derived gene for the 37 kDa secreted glycoprotein M-T7 gene was inserted and M-T7 expressed in CHO cells. Secreted M-T7 protein was isolated from concentrated, diafiltrated media (0.22 μM filter) and loaded on a QFF 40 mL column using a peristaltic pump with UV monitoring during FPLC. After loading, the column was washed with 25 mM Tris-HCl (pH7.4), 25 mM NaCl buffer followed by 25 mM Tris-HCl (pH 7.4), 1 M NaCl (with graded washes from 5% up to 100%). Contaminants were removed by thiophilic binding mode using 3.6 M (NH_4_)_2_SO_4_, 20 mM Hepes, pH7.0 added slowly to QFF40 25%B eluate to a final concentration of (NH_4_)_2_SO_4_ of 0.7 M. After filtration through a 0.22 μM filter, the eluate was loaded on a thiophilic column (8 mL), washed with 0.7 M (NH_4_)_2_SO_4_, 20 mM Hepes and M-T7 eluted by 20 mM Hepes, pH 7.0. M-T7 protein was dialyzed and further purified on a ceramic hydroxyapatite (CHT) column followed by dialysis against phosphate-buffered saline (PBS). M-T7 isolates were 96% pure with a single band and dimers detected on SDS PAGE (Figure 9). Endotoxin was assessed and below the limit of detection (0.125 EU/mL or 0.231 EU/mg). Final protein preps were filter sterilized using a 0.22 micron filter and stored at 4 °C for use within 1 week or stored at −80 °C for long-term storage [7,8,9,37,38].

### 4.5. Antisense to N Deacetylase Sulfotransferase-1 (Ndst1) ASONdst1 Construct Design

An ASONdst1 construct was developed and chemically modified to improve potency and stability (developed by SY; Target Sequence NDST1 ASO: CTGCAACTTACTTTTA; control ASO: GGCCAATACGCCGTCA). The ASO^Ndst1^ was designed with molecular composition and quality analysis for maximum stability, activity and lack of toxicity. These modifications included a phosphorothioate (PS) backbone and ribose modification, replacing cytosine with 5-methylcytosine (Figure 10A).

These ASOs have a 3-10-3 design, which means the first three and last three sugar molecules were cET bridged nucleic acids and the middle 10 sugars were unmodified deoxyribose nucleotides. NDST1 ASOs were synthesized and screened in cell culture for in vitro activity. This lead ASO was the most actively tolerated ASO in mice. The control ASO was selected with the same chemistry but did not target any mouse gene. ASO^Ndst1^ significantly reduced the detected Ndst1 expression in normal mice. (Figure 10B). ASO^Ndst1^ was assessed for any adverse effects in normal mice, no significant adverse effects were detected (Figure 10C).

### 4.6. Histopathology and Immunohistochemistry

Grafts were assessed for histological analysis with routine hematoxylin and eosin (H&E) staining as well as immunohistochemical (IHC) staining. Total inflammatory cell infiltrate area and scar (fibrotic) areas were measured and normalized to the total renal allograft implant area [7,8,9,12,13,35,38]. The diameter of the inflammation and scar were similarly measured and normalized to total graft diameter. The numbers of histologically normal glomeruli were also counted in renal allografts. A histopathology score was developed based upon the classifications used in the Banff criteria with assigned scores ranging from 1+ up to a maximum of 4+ for detectable tubule or glomerular inflammation or scarring (fibrosis), providing a score for the histopathology associated with acute damage, inflammation and rejection.

IHC was performed for the detection of cell and rejection markers: F4/80 for macrophage; CD3 for nonspecific T lymphocytes; Ly6G for neutrophils; CD19 for B lymphocytes and C4d as a marker for transplant rejection. For IHC, the primary antibodies included the following: Anti-mouse C4d Cat: HP8033; Ra pAb to F4/80 ab100790; Rb pAb to CD3m ab 5690 and secondary antibody: Goat antiRat IgG2a Hrp conjugated Cat: A110-109P. Sections were examined using an Olympus BX51 microscope with 4×–100× objectives, a Prior ProScan II stage and Olympus DP74 CMOS camera and cellSens software analysis system. Staining for C4d was used as a secondary marker of rejection.

### 4.7. Statistical Analysis

Immunohistochemical analyses were read initially by MB, JY and AL and then read by an investigator (IRZ) blinded to treatment and mouse strain. Significance in each parameter was assessed by StatView version 5.0.1 (SAS Institute, Inc., Cary, NC, USA) using one-way analysis of variance (ANOVA) with Fischer’s LSD (Least Significant Difference) comparison, or a Student’s unpaired *t-*test. *p* < 0.05 was considered significant.

## Figures and Tables

**Figure 1 pathogens-11-00588-f001:**
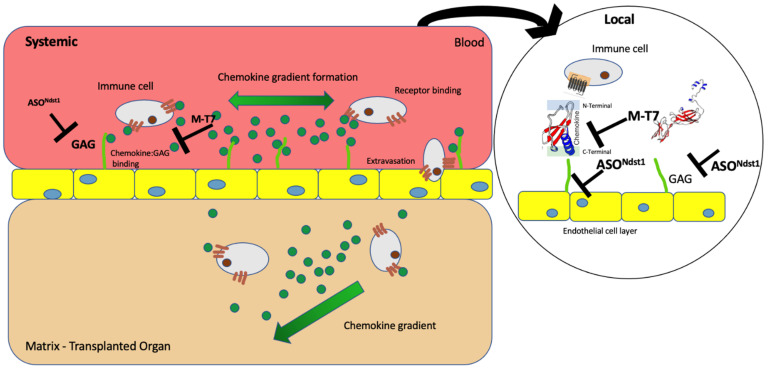
Chemokine GAG interaction–M-T7 and ASO^Ndst1^.

**Figure 2 pathogens-11-00588-f002:**
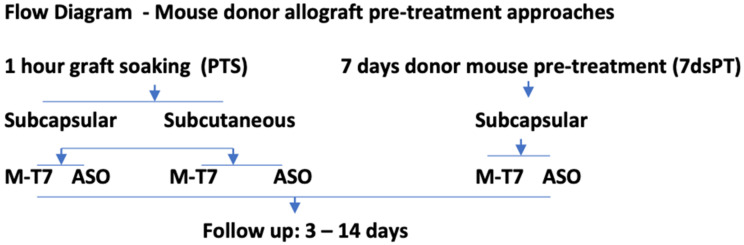
Flow Diagram of Transplant Studies.

**Figure 3 pathogens-11-00588-f003:**
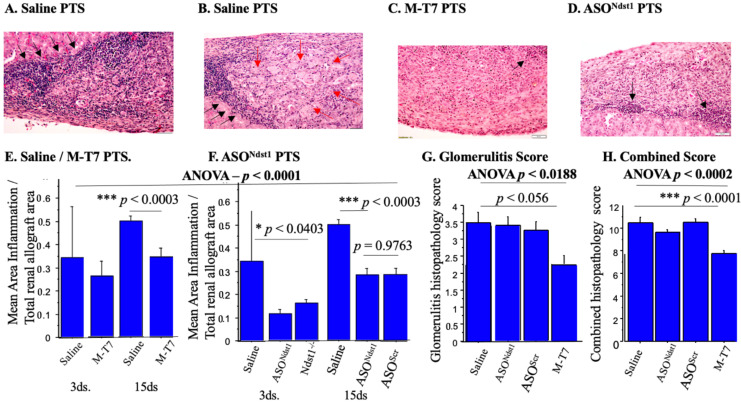
Inflammation in PTS pretreatment groups**.** Reduced inflammation is detected on H&E stained micrographs with M-T7 PTS pre-treatment of donor organs given immediately prior to engrafting. Black arrows indicate inflammatory cell invasion, red arrows highlight areas of scarring. (**A**). Subcapsular renal allografts pretreated with saline at 15 days follow-up, 20×; (**B**). Subcapsular renal allograft pretreated (PTS) with saline at 15 days follow-up, 40× (**B**); (**C**). PTS with M-T7 at 15 days follow-up showing decreased inflammation in comparison to saline, 40×; (**D**). (PTS) with ASO^Ndst1^ at 15 days follow-up, 40×. Black arrows indicating inflammation; (**E**). Bar graphs comparing ratios of mean area of inflammation/ total renal allograft area for saline versus M-T7 PTS allografts at 3 and 15 days. M-T7 significantly decreased area of inflammation in comparison to saline at 15 days follow-up (*** *p* < 0.0001 ANOVA, *** *p* < 0.0003); (**F**). Bar graphs comparing ratios of mean area of inflammation/total renal allograft area days (*** *p* < 0.0001 ANOVA; *p* < 0.0403) and at 15 days (*** *p* < 0.0003) in comparison to saline. ASO^Ndst1^ effects were equal to the control ASO^Scr^ (*p* = 0.9763) suggesting a non-specific effect for ASO^Ndst1^ on inflammation in PTS treated grafts; (**G**). Bar graphs comparing glomerulitis histopathology score of M-T7 PTS renal allografts versus controls. Glomerulitis score was significantly reduced with M-T7 treatment (*** *p* < 0.0188 ANOVA) at 15 days follow-up; (**H**). Bar graphs showing decreased combined histopathology score with M-T7 PTS renal allografts at 15 days follow-up in comparison to controls (*** *p* < 0.0002 ANOVA). * *p* < 0.05, *** *p* < 0.001.

**Figure 4 pathogens-11-00588-f004:**
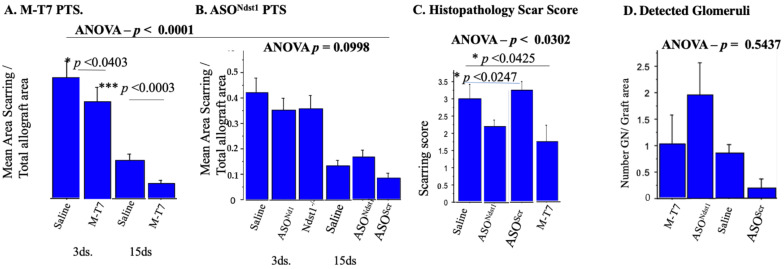
Scarring in PTS pretreatment groups. Analysis of scarring in PTS groups. (**A**). Bar graphs for mean area of Scarring/Total allograft Area at Subcapsular Renal allografts pre-treated (PTS) with MT7 in comparison to Saline. Reduction in the area of scarring was detected at both 3 and 15 days follow-up (*p <* 0.0001 ANOVA; *** *p* < 0.0003) with M-T7 treatment; (**B**). Bar graphs for Ratio of mean Area of Scarring/Total Allograft Area at Subcapsular Renal Allografts pre-treated (PTS) with ASO^Ndst1^. No significant decrease was found for measured scar area at both 3 and 15 days follow-up; (**C**). Bar graphs for independent Histopathology Scar Score at Subcapsular Renal Allografts PTS at 15 days follow-up. M-T7 has significantly reduced area of scarring (*p* < 0.0302 ANOVA). ASO^Ndst1^ has also demonstrated reduction on histopathology score for scarring (*p* < 0.0425); (**D**). Bar graphs for number of Detected Glomeruli/Total Graft area with PTS of Subcapsular Renal Allografts at 15 days follow-up. A trend toward an increased number of detected glomeruli with intact morphometry was found with M-T7 PTS treatments but significance was not reached (*p* < 0.5437 ANOVA). * *p* < 0.05, *** *p* < 0.001.

**Figure 5 pathogens-11-00588-f005:**
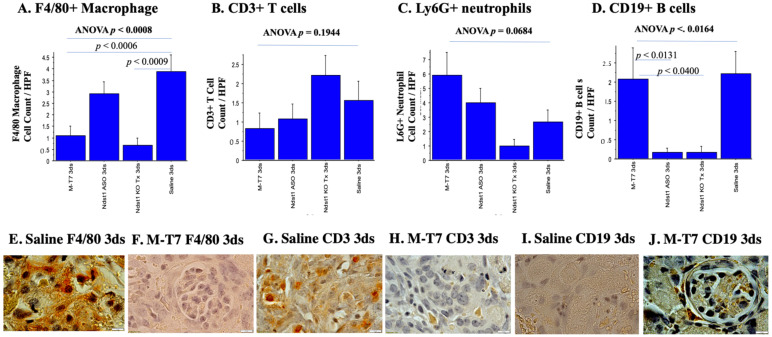
Immunohistochemistry demonstrates M-T7 reduces macrophage invasion in the immediate pre-treated (PTS) model. (**A**). F4/80+ Macrophage infiltrates were significantly reduced with M-T7 at 3 days follow-up (*p* < 0.0008 ANOVA). Reductions in this model were comparable to the ones seen in the Ndst1^−/−^ kidney transplants when compared to saline (*p* < 0.0009). ASO^Ndst1^, in contrast, did not alter macrophage invasion (*p* = 0.2007); (**B**). CD3+ T cell counts were not significantly altered by any of the pre-treatments at 3 days follow-up (*p*= 0.1944 ANOVA); (**C**). LyG6+ neutrophil cell counts at 3 days follow-up were not significantly altered by pre-treatments (*p* = 0.0684 ANOVA). Ndst1^−/−^ subcapsular transplants had reduced neutrophil counts when compared to saline, while M-T7 and ASO^Ndst1^ trended toward a non-significant increase; (**D**). CD19+ B cell counts were significantly decreased (*p* < 0.0164 ANOVA) by ASO^Ndst1^ (*p* < 0.0131) and Ndst1^−/−^ (*p* < 0.0400) pre-treatment of C57BL/6 mice at 3 days follow-up. M-T7 and saline treatments had equivalent effects on CD19+ B cell counts indicating neither suppression of nor increase in B cells by M-T7. Micrographs of pre-treated (PTS) subcapsular renal allografts with: (**E**). Saline PTS at 3 days follow-up showing F4/80+ macrophages, 100×; (**F**). M-T7 PTS at 3 days follow-up showing decreased F4/80+ macrophages, 100×. Glomeruli visualized inside the transplant; (**G**). Saline PTS at 3 days follow-up showing CD3+ T cells, 100×; (**H**). CD3+ T cells in M-T7 PTS at 3 days follow-up, 100×; (**I**). IHC for CD19+ B cells in Saline PTS at 3 days follow-up, 100×; (**J**). IHC with CD19+ B cells in M-T7 PTS at 3 days follow-up, histology micrograph at 100× shows intact glomeruli.

**Figure 6 pathogens-11-00588-f006:**
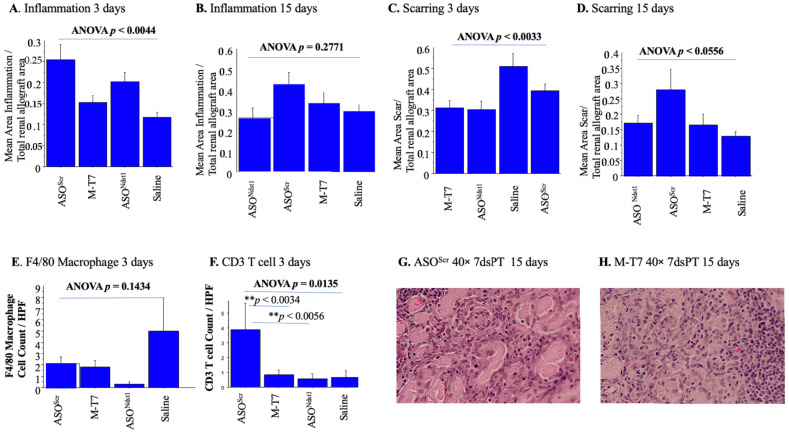
Inflammation and scarring after 7dsPT. Seven days pre-treatment of donor mice with M-T7 did not reduce inflammation, but reduced scarring. (**A**). Bar graphs of 7 days pre-treatment of Subcapsular Renal Allografts comparing Mean area of inflammation/Total allograft area at 3 days follow-up. ASO^Scr^ and ASO^Ndst1^ increased areas of inflammation when compared to Saline (*p* < 0.0044 ANOVA). M-T7 did not reduce inflammation when compared to saline; (**B**). Bar graphs of 7 days pre-treatment of Subcapsular Renal Allografts comparing Mean area of inflammation/Total allograft area at 15 days follow-up. There was a trend towards increased inflammation with ASO^Ndst1^ (*p* = 0.2771 ANOVA), but no significant change with any treatment; (**C**). Bar graphs for Mean area of Scar/Total Renal Allograft area at 3 days follow-up (*p* < 0.0033 ANOVA). Scarring was significantly reduced by 7dsPT with M-T7 (*p* < 0.0010), and by ASO^Ndst1^ 7dsPT (*p* = 0.13144), when compared to saline or ASO^Scr^ controls; (**D**). Bar graphs for Mean area of Scar/Total Renal Allograft area at 15 days follow-up (*p* < 0.0556 ANOVA). There was no overall decrease with 7dsPT with either ASO^Ndst1^ (*p* = 0.3690) or M-T7 (*p* = 0.4517) treatments when compared to saline. ASO^Scr^ treatment significantly increased scarring in comparison to ASO^Ndst1^ (*p* < 0.0479); (**E**). Bar graphs for F4/80 macrophages cell count of 7dsPT of Subcapsular Renal allograft at 3 days follow-up. M-T7 and ASO^Ndst1^ had nonsignificant trends toward reducing F4/80+ cell counts (*p* = 0.1434 ANOVA); (**F**). Bar graphs for CD3+ T cell count of 7dsPT of Subcapsular Renal allograft at 3 days follow-up. ASO^Scr^ increased CD3+ T cell counts when compared to Saline, M-T7 or ASO^Ndst1^ pre-treatments (*p* = 0.0135 ANOVA). (**G**). H&E Histology micrograph of Subcapsular Renal Allograft after 7dsPT with ASO^Scr^ at 15 days follow-up showing increased area of scarring, 40×; (**H**). H&E Histology micrograph of Subcapsular Renal Allograft after 7dsPT with M-T7 at 15 days follow-up showing decreased area of scarring, 40×. ** *p* < 0.01.

**Figure 7 pathogens-11-00588-f007:**
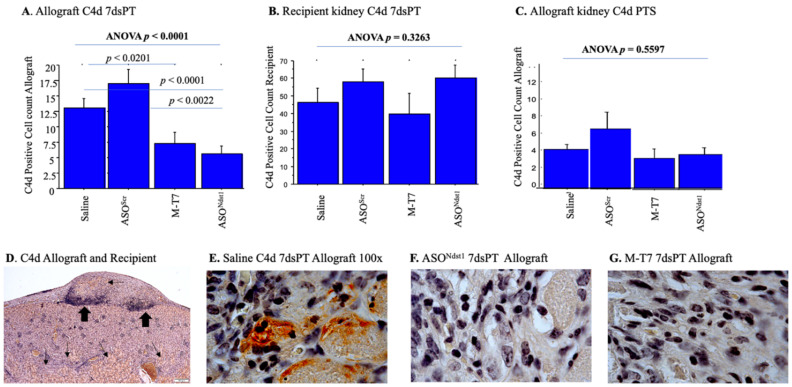
Immunohistochemical analysis C4d staining Allografts and Recipients- 7dsPT and PTS. Immunohistochemical analysis of C4d within Subcapsular Renal Allografts and the recipient Kidney with both 7dsPT and PT approaches. Areas of dense C4d positive tubules were detected in both models. (**A**). Bar graphs for C4d positive cell counts inside renal allograft sections with 7 days pre-treatment at 15 days follow-up (*p* < 0.0001 ANOVA). Both M-T7 (*p* < 0.0201) and ASO^Ndst1^ (*p* < 0.0001) significantly decreased C4d positive staining in comparison to ASO^Scr^ and Saline treatment controls; (**B**). Bar graphs for C4d positive cell count in the recipient kidney pretreated for 7 days at 15 days follow-up. C4d staining was not altered by any treatment (*p* = 0.3263 ANOVA); (**C**). Bar graphs for C4d positive cell count at Subcapsular Renal Allograft with PTS at 15 days follow-up. Increase in C4d staining was not altered by any of the treatments (*p* = 0.5597 ANOVA); (**D**). Histology micrograph of Subcapsular Renal allograft and Recipient kidney with immunohistochemistry pretreated for 7 days with saline at 15 days follow-up, 4×. Small black arrows point to areas of dense C4d positive staining; big black arrows point to inflammatory cells. Histology micrographs of subcapsular renal allograft pretreated for 7 days at 15 days follow-up (7dsPT) with: (**E**). saline. IHC showing C4d positive cells, 100×; (**F**). ASO^Scr^, 100×; (**G**). M-T7, 100×.

**Figure 8 pathogens-11-00588-f008:**
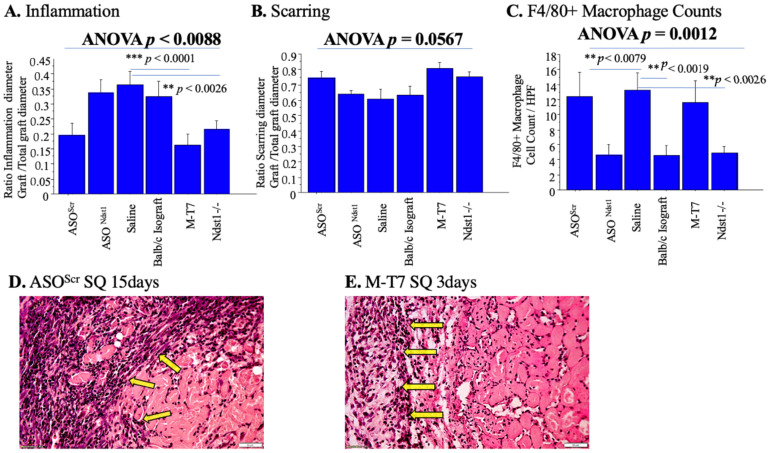
Subcutaneous Allograft Transplant–PTS treatment pre-engrafting. Analysis of Results for the pre-treatment (PTS) of Subcutaneous Renal Allograft Transplant Model. Pretreatment with M-T7 has significantly decreased inflammation and scarring. (**A**). Bar graphs comparing ratio of inflammation diameter/total graft diameter for the different pre-treatments of subcutaneous allograft at 15 days follow-up. PTS with M-T7 significantly reduced inflammation (M-T7 *p* < 0.0001, *p* < 0.0088 ANOVA). Pretreatment in Ndst1^−/−^ deficient mouse renal transplants significantly reduced inflammation; (**B**). Bar graphs comparing ratio of scarring diameter/total graft diameter for the different pre-treatments of Subcutaneous allograft at 15 days follow. None of the treatments reduced scarring in the subcutaneous transplant model (*p* = 0.0567 ANOVA); (**C**). Bar graphs comparing F4/80+ macrophage cell count for the different pre-treatments of subcutaneous allografts at 15 days follow-up (*p =* 0.0012 ANOVA). ASO^Ndst1^ has significantly decreased macrophage count in comparison to saline and ASO^Scr^ (*p* < 0.0079). F4/80+ Cell count was also decreased in Ndst1 deficient mouse renal transplants (*p* < 0.0026); (**D**). H&E Histology micrograph of Subcutaneous Renal Allograft pretreated (PTS) with ASO^Scr^ at 15 days follow-up. Yellow arrows pointing to area of dense inflammation at the transplant, 20×; (**E**). H&E Histology micrograph of Subcutaneous Renal Allograft pretreated (PTS) with M-T7 at 3 days follow-up. Yellow arrows pointing to inflammatory cells, 20×. ** *p* < 0.01, *** *p* < 0.001.

**Figure 9 pathogens-11-00588-f009:**
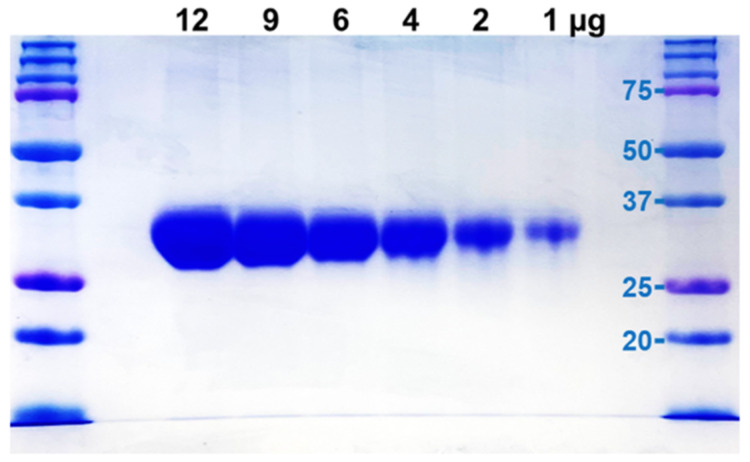
M-T7—Gel electrophoresis illustrating M-T7 purification. Monomeric M-T7 bands are present at approximately 35 kD with evidence for dimers at 70 kD.

**Figure 10 pathogens-11-00588-f010:**
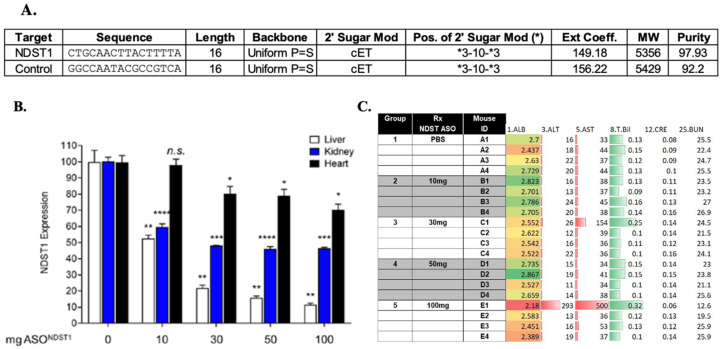
ASO^Ndst1^ sequence and activity. (**A**). Sequences for ASO^Ndst1^ and ASO^Scr^ Control; (**B**). Significantly reduced Ndst1 expression is detected in normal, non-transplanted C57BL/6J mice treated with ASO^Ndst1^ demonstrated reduced Ndst1 expression on Taqman RT-qPCR. A greater percentage reduction in Ndst1 expression is seen after ASO^Ndst1^ treatment at doses above 30 mg; (**C**). Doses at 10–50 mg are entirely safe in normal mice without transplant. AST and bilirubin were increased at doses of 100 mg. Two-Way ANOVA w/Benjamini post-hoc vs. 0 mg. * *p* < 0.05, ** *p* < 0.01, *** *p* < 0.001, **** *p* < 0.0001.

**Table 1 pathogens-11-00588-t001:** Renal allograft models; treatments and allograft numbers.

Renal Allograft Pretreatment-PTS and 7dsPT	Transplant Donor Mouse–Recipient Mouse	Treatment	Days Follow Up	Number of Transplant Procedures
**PTS** **Subcapsular transplant Pre-treatment Soak-1 h**	C57BL/6-BALB/c	Saline	3 days	4
	Ndst1^−/−^ C57Bl/6-BALB/c	Saline		4
	C57BL/6-BALB/c	M-T7		4
	C57BL/6-BALB/c	ASO^Ndst1^		4
	C57BL/6-BALB/c	ASO^Scr^		4
**PTS** **Subcapsular transplant Pre-treatment Soak-1 h**	C57BL/6-BALB/c	Saline	15 days	10
	Ndst1^−/−^ C57Bl/6-BALB/c	Saline		4
	C57BL/6-BALB/c	M-T7		5
	C57BL/6-BALB/c	ASO^Ndst1^		7
	C57BL/6-BALB/c	ASO^Scr^		4
**7dsPT Subcapsular transplant-** **Pre-treatment donor-7 days**	C57BL/6-BALB/c	Saline	3 days	6
	C57BL/6-BALB/c	M-T7		6
	C57BL/6-BALB/c	ASO^Ndst1^		6
	C57BL/6-BALB/c	ASO^Scr^		6
**7dsPT Subcapsular transplant-7days Pre-treatment donor-7 days**	C57BL/6-BALB/c	Saline	15 days	6
	C57BL/6-BALB/c	M-T7		6
	C57BL/6-BALB/c	ASO^Ndst1^		6
	C57BL/6-BALB/c	ASO^Scr^		6
**PTS** **Subcutaneous transplant Pre-treatment Soak-1 h**	C57BL/6-BALB/c	Saline	3 days	6
	Ndst1^−/−^ C57Bl/6-BALB/c	Saline		6
	C57BL/6-BALB/c	M-T7		3
	C57BL/6-BALB/c	ASO^Ndst1^		6
	C57BL/6-BALB/c	ASO^Scr^		6
	BALB/c-BALB/c	Saline		5
**PTS** **Subcutaneous transplant Pre-treatment Soak-1 h**	C57BL/6-BALB/c	Saline	15 days	2
	Ndst1^−/−^ C57Bl/6-BALB/c	Saline		6
	C57BL/6-BALB/c	M-T7		4
	C57BL/6-BALB/c	ASO^Ndst1^		8
	C57BL/6-BALB/c	ASO^Scr^		4
	BALB/c-BALB/c	Saline		6

PTS—pre-treatment of donor organ with flush and soaking X 1 h, 7dsPT-7 days pre-treatment of donor mouse, Ndst1^−/−^—Ndst1 conditional knockout mouse, ASO^Ndst1^—antisense to Ndst1; ASO^Scr^—antisense control with scrambled sequence.

## Data Availability

Not applicable.

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
