# Peer review of "Virus-Derived Chemokine Modulating Protein Pre-Treatment Blocks Chemokine–Glycosaminoglycan Interactions and Significantly Reduces Transplant Immune Damage"

_pathogens, 2022, doi:10.3390/pathogens11050588_

Round 1

Reviewer 1 Report

The article submitted for review is very interesting and, in my opinion, it is based on reliable research. I consider the choice of the topic to be interesting and important in the clinical context. It has been known for a long time that the invasion of immune cells after solid organ transplantation, in this case, the kidney, is directed by the binding of chemokines to glycosaminoglycans, creating gradients directing the infiltration of immune cells. This is a huge challenge in transplantology, especially in the context of the most common transplants - kidney transplants - the most common due to the prevalence of pathologies and organ availability. The authors of the study looked at this phenomenon and followed the strategy of viruses that developed highly active chemokine inhibitors as a way to avoid host response. The M-T7 protein from myxomavirus blocks the chemokine: GAG binding. Researchers investigated the M-T7 protein as well as antisense (ASO) as a pretreatment to modify the chemokine: GAG interaction to reduce damage to donor organs. As a result of the analyzes, the authors of the study observed that immediate pretreatment of donor's kidneys with M-T7 to block the chemokine: GAG binding significantly reduced inflammation and scarring in allografts. ASONdst1, on the other hand, was less effective with direct pretreatment, but reduced scarring and C4d staining with donor pretreatment for 7 days prior to transplantation. Interestingly, the researchers found that the transplants with conditional Ndst1 deficiency had reduced inflammation. I believe that the researchers have made a valid conclusion from the study that local chemokine inhibition: GAG binding in donor organs immediately prior to transplantation provides a new approach to reducing graft damage and graft loss.

It would be helpful to draw a diagram that would allow the reader to follow the following stages of the research.

Additionally, I recommend drawing a diagram including chemokine binding to glycosaminoglycans (GAGs) in the context of immune cell infiltration in the transplanted organ/kidney. It will help the readers to understand the studied phenomenon more deeply.

Another issue is stylistic errors. I believe the authors should reread the article and correct some phrases.

Reviewer 2 Report

I considered the manuscript entitled “Virus-derived Chemokine Modulating Protein Pre-treatment Blocks Chemokine-Glycosaminoglycan Interactions and Significantly Reduces Transplant Immune Damage” by Isabela R Zanetti that is intended to be published in Pathogens journal.

This is a highly comprehensive manuscript with several groups of animals and experimental scenarios. A tremendous effort from the authors. Globally, it is hard to follow and understand. At last, everything is understandable but, to me, readers will get tired and there is a possibility that they will abandon reading. There is a great amount of work which must be simplified in the exposition. Figures are excessively variegated and difficult to follow.

Major criticism is that in this non vascularized renal transplant model you aim to study changes in the interaction between endothelium and adaptive immunity response. Though you show a reduction in the degree of inflammation among groups, I am not sure you have achieved any demonstration of mechanisms postulated.

Introduction must be globally reduced, focused, and harmonized.

It should be introduced and discussed the non-heart beating donors as source of endothelial damage (Transplanted organs are damaged even prior to engrafting by ischemia due to low blood flow during transport or shock, surgical trauma and immunological damage induced by cytokine storm in donors with severe brain injury)

termed chronic rejection.Try to change this term, according to Banff classification

Increased expression of chemokines and chemokine receptors is detected in brain death. Where? Detail

Cellular activation and invasion into donor organs can be protective but, when excessive, cause progressive damage to donor grafts with scarring and graft loss: Non-sense in the paragraph

Repetitive?: While current immune modulating therapeutics are highly effective for acute antibody mediated rejection, many organs are damaged by this insidious, chronic and progressive immune rejection that causes vascular disease and scarring, termed chronic rejection.

Not necessary: While current immune modulating therapeutics are highly effective for 115 acute antibody mediated rejection, many organs are damaged by this insidious, chronic 116 and progressive immune rejection that causes vascular disease and scarring, termed 117 chronic rejection. Late or chronic rejection, that occurs after the first-year post transplant, 118 increases graft loss and treatment remains less effective necessitating repeat transplanta-119 tion in some patients or a return to hemodialysis. There is also a risk of medication in-120 duced toxicity [1,2,23]. High doses of immunosuppressants such as calcineurin inhibi-121 tors and corticosteroids can increase susceptibility to opportunistic infections amongst 122 other severe complications, including malignancies, diabetes and Cushing’s syndrome. 123 New drugs are now under investigation as approaches to reduce ths\is excess damaging 124 inflammation in donor grafts [23-31].

siRNA against which target?: Recent studies have investigated pre-136 treatment of liver and heart transplants with siRNA approaches in preclinical models

Repetitive: In summary, chemokine: GAG interactions are predicted to drive excess damaging 139 immune cell invasion in engrafted organs; damage produced by surgery, ischemia during 140 transport and cytokine storm in brain death.

Methods

It needs a better harmonization and clarification. A simpler and focused wording would be of interest for better understanding. The acronyms used to name the groups are hard to remember, and during the reading you have to continuously come back to the methods to understand what you are talking about

Repetitive, the Table is clear: Table 1; N = 50 subcapsular PTS renal 561 transplants, 20 with 3 days follow up and 28with 15 days follow up; N = 48 subcapsular 562 7dsPT renal transplants, 24 with 3 days follow up and 24 with 15 days follow up).

Explain better: Sections of 565 C57BL/6 mice kidneys were used as donors for BALB/c recipient mice; One C57BL/6 kid-566 ney divided into sections for implant into 6 recipient BALB/c mice [35].

It appears as the transplant procedure is not a vascularized model. How do you assess that the recipient cells and antibodies and cytokines reach the endothelium?

The description of the histological evaluated scores is not completely understandable. What is the composite score, the histopathologic scar score, the detected glomeruli……?

Results

The forest does not let you see the trees. Looking to the data and Figures from the distance you can appreciate differences between saline and the treatments. However, again, Figures are too variegated, and the text and Legends for Figures are too heavy to follow and reading.

Why C4d was positive also in the native kidney? It is not obvious, as native endothelium should appear as excluded from immunologic storm. Is the C4d staining reproducible in adequate negative and positive controls? Did you check this positivity in native kidneys coming from mouse other than those in the present experiments? The presented photomicrography is not the classical from this staining. Usually, the epithelium appears stained in a thin manner, and blood cells are clearly seen inside.

Discussion.

It should be shortened and more focused

Reviewer 3 Report

The authors investigated M-T7 M-T7 (a 37KDa myxomavirus-derived purified 70 protein that binds C, CC and CXC classes of chemokines blocking chemokine to GAG 71 binding) and also antisense (ASO) (construct targeting 144 Ndst1 (ASONdst1) prior to donor organ implantation.) as pre-treatments to modify chemokine: GAG interactions to reduce donor organ damage.

The results show that Immediate pre-treatment of donor kidneys with M-T7 to block chemokine: GAG binding significantly reduced inflammation and scarring in subcapsular and subcutaneous allografts. Antisense to N-deacetylase N- sulfotransferase1 35 (ASONdst1) that modifies heparan sulfate, was less effective with immediate pretreatment, but reduced scarring and C4d staining with donor pre-treatment for 7days before transplantation.

Grafts with conditional Ndst1 deficiency had reduced inflammation. Local inhibition of chemokine:

Thae athors conclude that GAG 38 binding in donor organs immediately prior to transplant provides a new approach to reduce transplant damage and graft loss.

Very interesting manuscript!!!

I would like to know the response of Dendritic cells  because a recently published article about this issue  ( https://doi.org/10.3389/fimmu.2021.654540) hit my attention

Round 2

Reviewer 2 Report

It is suitable for publication though all the manuscript could have beenreduced and simplified

Reviewer 3 Report

The authors answered correctly to my question